# Enhancing Cervical Cancer Prevention in South African Women: Primary HPV mRNA Screening with Different Genotype Combinations

**DOI:** 10.3390/cancers15225453

**Published:** 2023-11-17

**Authors:** Sveinung Wergeland Sørbye, Bente Marie Falang, Matthys H. Botha, Leon Cornelius Snyman, Haynes van der Merwe, Cathy Visser, Karin Richter, Greta Dreyer

**Affiliations:** 1Department of Clinical Pathology, University Hospital of North Norway, 9038 Tromsø, Norway; 2PreTect AS, 3490 Klokkarstua, Norway; bente.falang@pretect.no; 3Department of Obstetrics and Gynaecology, Faculty of Medicine and Health Sciences, Stellenbosch University, Cape Town 7505, South Africa; mhbotha@sun.ac.za (M.H.B.); haynes@sun.ac.za (H.v.d.M.); 4Gynaecological Oncology Unit, Department of Obstetrics and Gynaecology, Faculty of Health Sciences, University of Pretoria, Pretoria 0028, South Africa; leon.snyman@up.ac.za (L.C.S.); cathy.visser@up.ac.za (C.V.); gretadreyer@mweb.co.za (G.D.); 5Department of Medical Virology, Faculty of Health Sciences, University of Pretoria, Pretoria 0028, South Africa; karin.richter@lancet.co.za

**Keywords:** cervical cancer, HPV screening, prevention strategies, human papillomavirus, HIV interaction, “Test and Treat” approach, number needed to treat, positive predictive value, negative predictive value, risk management, global health challenges

## Abstract

**Simple Summary:**

Despite being preventable, cervical cancer remains a lead killer among women in low–middle income countries. The reasons are clear. Global inequity is reflected in limited health resources, such as limited access to HPV-vaccination, screening, and treatment, which are the key pillars for the elimination of cervical cancer. The high burden of HPV-coinfections in women living with HIV/AIDS challenges the use of sensitive HPV-testing, leading to massive over-treatment. As advised by the WHO, a “test-and-treat” strategy could be optimal. In this study, 710 under-screened South African women, about half of them living with HIV, were tested for mRNA expression restricted to the predominant HPV types of cervical cancer. The effectiveness of a test-and-treat strategy to detect and manage severe cervical abnormalities (CIN3+) by increasing the number of HPV types included in the test was modeled. A combination of 6 types (16, 18, 31, 33, 35, 45) resulted in 25% positive tests requiring treatment, with the potential to prevent up to 85% of all cervical cancers.

**Abstract:**

Background: Cervical cancer prevention in regions with limited access to screening and HPV vaccination necessitates innovative approaches. This study explored the potential of a test-and-treat strategy using mRNA HPV tests to impact cervical cancer prevention in a high-prevalence HIV population. Methods: A cervical screening study was conducted at three South African hospitals involving 710 under-screened, non-pregnant women (25 to 65 years) without known cervical diseases. Cytology, HPV testing, colposcopy, and biopsies were performed concurrently. Histopathologists determined final histological diagnoses based on biopsy and LLETZ histology. mRNA-HPV-genotyping for 3 (16, 18, 45) to 8 (16, 18, 31, 33, 35, 45, 52, 58) high-risk types was performed on leftover liquid-based cytology material. The preventive potential of the test-and-treat approach was estimated based on published data, reporting the causative HPV types in cervical cancer tissue from South African women. Treatment was provided as needed. Results: The HPV positivity rate more than doubled from 3-type (15.2%; 95% CI: 12.6–17.8) to 8-type mRNA (31.5%; 95% CI: 28.8–34.9) combinations, significantly higher among HIV-positive women. CIN3+ prevalence among HIV-positive women (26.4%) was double that of HIV-negative women (12.9%) (*p* < 0.01). The 6-type combination showed the best balance of sensitivity, specificity and treatment group size, and effectiveness to prevent cervical cancer. A 4-type combination (16, 18, 35, 45) could potentially prevent 77.6% (95% CI: 71.2–84.0) of cervical cancer burden by treating 20% and detecting 41.1% of CIN3 cases in the study group. Similarly, a 6-type combination (16, 18, 31, 33, 35, 45), treating 25% and including 62% of CIN3 cases, might prevent 85% of cervical cancer cases (95% CI: 79.6–90.6) among HIV-positive and negative women. Conclusion: Employing mRNA HPV tests within a test-and-treat approach holds huge promise for targeted cervical cancer prevention in under-screened populations. Testing for mRNA of the 6 highest-risk HPV types in this population and treating them all is projected to effectively prevent progression from CIN3 to invasive cervical cancer while reducing overtreatment in resource-constrained settings.

## 1. Introduction

Today’s extensive knowledge and applied scientific advancements regarding human papillomavirus (HPV) as the pivotal causative agent of cervical cancer offer profound opportunities for enhancing cervical cancer prevention strategies. Cervical cancer screening programs have evolved significantly since the 1960s, progressing from opportunistic to interval screening, and more recently adopting molecular tests targeting oncogenic HPV types. However, these advancements pose new challenges, including limited healthcare resources, how to manage screen positives and structural impediments best, selecting the appropriate test systems, determining HPV types to include, establishing screening intervals, devising follow-up and treatment algorithms, and defining target populations [1,2,3,4].

The World Health Organization (WHO) envisions cervical cancer as a preventable ailment. Their elimination strategy entails 90% vaccine uptake, and screening coverage of 70% using a high-performance HPV test followed by treating 90% of screen-positive women, to achieve the ambitious goal of an incidence rate of less than 4 per 100,000 women in all countries [5]. Obviously, for low- and middle-income countries bearing the highest burden of cervical cancer, global political commitment and support are necessary for implementing such initiatives. In such resource-constrained regions, the implementation of national programs has been subject to substantial variation, where women have limited access to screening, vaccination, and treatment [6,7].

The South African Guidelines for Cervical Cancer Screening were introduced to provide asymptomatic women aged 30 years and older with three free Pap smears at 10-year intervals throughout their lifetime. However, the ambitious goal of screening at least 70% of women nationally within a decade of its initiation in 1999 has faced significant challenges, and a suboptimal coverage below 50% is reported [8,9]. The scarcity of healthcare personnel, inadequate equipment and facilities, the issue of patients lost to follow-up while awaiting colposcopy and treatment after abnormal cytology findings, and the absence of effective awareness programs may collectively contribute to the lack of success of this screening initiative [10]. By 2021, HPV testing was primarily research-based, and performed in selected private laboratories [9,11].

The school-based HPV vaccination initiative using a two-dose schedule (targeting HPV types 16 and 18) commenced in 2014, is focusing on girls aged nine and above and reporting first dose coverage of 69% in 2019 [7,12,13]. South Africa reports one of the highest age-standardized cervical cancer incidence rates globally (35.3 per 100,000 women years in 2020), notably driven by the interplay between human immunodeficiency virus (HIV) and HPV infections, compounded by the suboptimal efficacy of the existing screening program [14,15]. With nearly eight million people affected by a HIV infection, South Africa has a significant population of women living with HIV (WLWH) who face a 6.1-fold higher relative risk of cervical cancer compared to the general population [16,17].

To formulate an optimal screening and vaccination strategy to reduce the burden of cervical cancer in South Africa, accurate data on HPV prevalence and types, stratified by HIV status, are essential. The understanding that type-specific HPV prevalence varies by geography and cervical abnormalities, underscores the importance of determining which HPV types to include in primary and secondary prevention [14,18,19]. Most countries with HPV screening have chosen a 14-type HPV DNA test, based on the association of these types with CIN2-3 lesions, irrespective of the regional distribution of HPV types identified in cervical cancer [20]. Evidently, the risk of future cellular anomalies and eventually invasive cancer is attributed to the heightened expression of the E6/E7 viral oncogenes, not the infection per se [21,22]. A significant proportion of HPV infections are transient and naturally cleared by the immune system within 1 to 2 years, resulting in a considerable risk of overdiagnosis and overtreatment among women who test positive for HPV during primary screening [23]. Moreover, it is firmly established that E6/E7 mRNA biomarkers exhibit a robust correlation with the risk of cervical cancer progression [24,25,26,27,28,29,30,31]. The detection of mRNA E6/E7 has shown higher specificity in detecting clinically relevant disease CIN2+ compared to conventional HPV-DNA testing [32,33,34,35]. In the context of the high HPV prevalence in South Africa, the ambitious treatment rates suggested by the WHO necessitate a test-and-treat approach [5].

This study aimed to evaluate the efficacy of HPV mRNA tests in primary cervical cancer screening, assessing various combinations of mRNA HPV types to predict histologically proven CIN3+ lesions in women based on their HIV status. Additionally, the study modeled the effectiveness of a test-and-treat approach in detecting and managing CIN3+ lesions, with a focus on optimizing the selection of HPV types for screening. The analysis included an assessment of the number mRNA positive cases that needed to be treated in order to address one CIN3+ case as a measure of screening effectiveness. The choice of HPV types for the screening algorithm was informed by the HPV types identified in invasive cancer cases from the region. This approach aims to prevent the progression of CIN3 to invasive cervical cancer, particularly in resource-constrained settings where access to colposcopy, biopsy, and pathologists may be limited.

## 2. Material and Methods

### 2.1. Study Population and Recruitment

Between December 2016 and March 2020, the multicentric DIAgnosis in Vaccine and Cervical Cancer Screen (DiaVACCS) screening trial, enrolled a total of 1104 women aged 25 to 65 years from general and anti-retroviral outpatient clinics at two hospitals in Pretoria and one hospital in Cape Town, South Africa [36]. Participants were specifically selected if they had not received any cervical screening results within the past five years and provided informed consent to undergo study procedures. Given the age range of the study population and the focus on unscreened women, it is important to note that the vast majority of young women included in the study were unlikely to have received the HPV vaccine. Excluded from participation were pregnant women, individuals with prior hysterectomy, current or previous gynecologic cancer, or known pre-cancerous conditions. Women who did not consent or were otherwise unable to undergo screening or treatment if indicated, were not enrolled. The study sites were fully equipped for screening, testing, and treatment, as well as appropriate cold storage facilities. Clinical research staff were trained to administer study protocols, with specialist oversight readily available. All medical visits, tests, and treatments were provided free of charge, without compensation to participants. Population demographics and baseline results for visual inspection, cytology, and HPV DNA prevalence of phase 1 of the DiaVACCS screening trial have been published [36]. This current report included 1001 participants who were part of the larger DiaVACCS screening trial and for whom residual liquid-based cytology (LBC) samples were available for HPV mRNA testing. To ensure the robustness of the study, 274 women lacking biopsy and histological evaluation, one case with endometrial cancer, and 16 women with either invalid mRNA test results or unknown HIV status, were excluded, resulting in data from 710 women being eligible for statistical analysis (Figure 1). The exclusion of cases without histology data resulted in a study population enriched for cervical disease in comparison to the original screening population. This is important due to a relatively large number of screen-negative women who had positive histology.

### 2.2. Study Protocol and Procedures

The study protocol dictated that any woman presenting with abnormalities detected during visual inspection, HPV testing, or cytology tests would be scheduled for a follow-up colposcopy and biopsy. If a lesion was observed, targeted biopsies were taken, and in those without, at the 6 o’clock and 12 o’clock positions. All women with positive screening tests and a cohort of screen negatives were recalled for treatment or biopsy. In response to low attendance rates for this subsequent visit, the protocol was adjusted early in the study to conduct a colposcopy and biopsy during the initial visit. Treatment, when necessary, involved large loop excision of the transformation zone (LLETZ) and was performed as an outpatient procedure. Histology was eventually available for 91.7% (555/605) of screen-positive participants (605/1104) and 42.7% (213/499) of screen-negatives (499/1104) [36].

### 2.3. Diagnostic Evaluation and Classification

The Bethesda system served as the framework for reporting and classifying cervical cytology. All cervical biopsies underwent routine microscopic examination and processing. An experienced pathologist following the World Health Organization’s Cervical Intraepithelial Neoplasia (CIN) classification [37] evaluated the biopsies to determine the appropriate follow-up steps. The final histology result was determined based on the most severe finding among punch biopsies and LLETZ specimens.

### 2.4. HIV Status and Categorization

For participants with undisclosed or unknown HIV status, a rapid HIV test (Advanced Quality Rapid Anti-HIV (1&2) Test, InTec PRODUCTS, INC., Xiamen, China) was administered. HIV-positive participants provided information on their antiretroviral (ARV) therapy usage. Age was categorized into two groups (25–39 years and 40–65 years), while HIV status, ARV therapy, and HPV mRNA types were classified into three categories (invalid, negative, or positive). ARV status is only reported in Phase 1 baseline results, and not stratified within the data reported here.

### 2.5. HPV mRNA E6/E7 Testing

HPV mRNA E6/E7 expression was assessed using two commercially available tests: PreTect SEE-SA and PreTect HPV-Proofer’7 (PreTect AS, Klokkarstua, Norway). These tests enabled the detection and direct genotyping of high-risk HPV genotypes, including four (16, 18, 45, 35) and seven (16, 18, 31, 33, 45, 52, 58) genotypes. The qualitative assays employed real-time nucleic acid sequence-based amplification (NASBA) technology, targeting full-length E6/E7 transcripts and incorporating an intrinsic sample control for sample adequacy. The assay results were validated using positive and negative controls, which corresponded to viral mRNA targets for all specified types. Total nucleic acids were extracted from 1 mL of the residual frozen liquid-based cytology (LBC) material preserved in ThinPrep, using PreTect X, and subsequently analyzed for mRNA expression following the manufacturer’s instructions.

### 2.6. Analysis of mRNA Assays and Cervical Cancer Prevention Potential

The results from the two mRNA assays were utilized to evaluate test performance for detecting CIN3 disease and cervical cancer. Various combinations of HPV types, ranging from three (16, 18, 45) to eight (16, 18, 45, 31, 33, 52, 58, 35), were assessed, and the subsequent impact on cervical cancer prevention was estimated based on the reported types detected in cervical cancer cases from the same hospitals [38]. The prevalence data presented by Rad et al. allowed the estimation of the potential contribution of each HPV type to the development of cervical cancer. Based on the proportion of cases caused by different HPV types, a calculation of the projected cervical cancer prevention potential was made. This enabled us to predict the number of cervical cancer cases that could be prevented by treating precancerous lesions (CIN3+) caused by specific HPV types. We used this approach to assess the potential impact of including different HPV types in our screening test strategy. We acknowledge that this estimation involved certain assumptions and was based on limited available data, and therefore, the actual outcomes may vary. However, the approach allowed us to explore the trade-off between overtreatment of precancerous lesions and cervical cancer prevention. The eight selected HPV types in this study represented the major etiological types diagnosed in cervical cancer globally [39,40].

### 2.7. Statistical Analysis

The data were analyzed using Statistical Package for Social Sciences version 29.0. Armonk, NY, USA: IBM Corp. The significance of associations was assessed using the Chi-square test and the Chi-square test for trend, with a significance level set at *p*-values < 0.05. Confidence intervals for percentages and proportions, such as test performance outcomes and the estimated number of prevented cervical cancers, were calculated as the value of the percentage or proportion plus or minus the standard error, multiplied by 1.96.

### 2.8. Ethical Considerations

The study protocol received approval from the Faculties of Health Sciences Research Ethics Committee at the University of Pretoria, Pretoria, under reference number 196/2014 and Stellenbosch University, reciprocal approval 2015, and was registered as a clinical trial (ClinicalTrials.gov identification number NCT02956031). All study participants provided written informed consent prior to their inclusion in the study.

## 3. Results

### 3.1. Characteristics and Positivity Rate

Age distribution showed a similarity between the two HIV groups. Overall HPV mRNA positivity rate was 31.5% (224/710), where the most common types were found to be HPV 16, 45, 58, and 18 in decrescent order (Table 1). The HPV mRNA positivity rate surged from 15.2% (95% CI: 12.6–17.8) for the 3-type mRNA combination to 31.5% (95% CI: 28.8–34.9) for the 8-type mRNA combination, consistently demonstrating 2–3-fold increases and notably higher rates among HIV-positive compared to HIV-negative women for all genotype combinations (Table 2). The prevalence of CIN3+ among HIV-positive women (26.4%) was twice that of HIV-negative women (12.9%) (*p* < 0.01) (Table 2).

### 3.2. HPV Prevalence and Histological Diagnoses

Across all histological diagnoses, the prevalence of HPV on various mRNA combinations was higher among HIV-positive women compared to HIV-negative women but not statistically significant (overlapping confidence intervals not shown) (Table 3). Three out of four cervical cancers in HIV-negative women were positive for mRNA types 16, 18, and 45 but one case was negative for all HPV types tested for in this study. All eight cervical cancers in HIV-positive women exhibited mRNA positivity for types 16, 18, 45, 31, and 33.

### 3.3. Performance of HPV mRNA Assays for CIN3+ Detection

Sensitivity for CIN3+ disease showed a significant increase from screening with three mRNA types to eight mRNA types (Chi-square trend *p* < 0.001). The sensitivity rates were 31.3% (95% CI: 26.6–36.0) and 66.6% (95% CI: 61.8–71.4) among HIV-negative women, and 46.1% (95% CI: 10.8–51.4) and 84.3% (95% CI: 80.4–88.2) among HIV-positive women, for three and eight mRNA types, respectively (Chi-square trend *p* < 0.001) (Table 4). Similarly, specificity decreased significantly with an increasing number of HPV types, with a milder decrease observed among HIV-negative women (Chi-square trend = 12.9; *p* < 0.001) compared to HIV-positive women (Chi-square trend = 27.1; *p* < 0.001).

The positive predictive value (PPV) of various mRNA combinations for CIN3+ varied from less than 50% in HIV-negative women (Chi-square trend *p* = 0.92) to slightly above 50% for HIV-positive women (Chi-square trend *p* = 0.45), with no substantial difference across HIV status. The negative predictive value increased as the number of included mRNA types rose in the exposure categories, more for HIV-positive women (Chi-square trend = 13.5; *p* < 0.001) compared to HIV-negative women (Chi-square trend = 7.1; *p* < 0.01) (Table 4).

### 3.4. Impact on Cervical Cancer Prevention

In a prior investigation by Rad et al., the distribution of HPV types was examined in 159 women with confirmed cervical cancer and known HIV status. This cohort was drawn from the same hospitals involved in our present study [38]. Leveraging the reported South African prevalence data, we assessed the potential impact of various combinations of mRNA HPV types in primary cervical cancer screening.

In elucidating the potential impact of mRNA HPV screening, it is crucial to acknowledge the nuanced distribution of HPV types in cervical intraepithelial neoplasia grade 3 (CIN3) as opposed to invasive cervical cancer. CIN3 may exhibit a diverse range of HPV types, reflecting the heterogeneity in precancerous lesions. In contrast, our analysis of cervical cancer cases in South Africa revealed a substantial concentration, with 85% attributed to HPV types 16, 18, 45, 31, 33, and 35. While CIN3 may manifest with various HPV types, our focus on the high-risk types prevalent in invasive cervical cancer underscores the targeted approach of our screening strategies. Detecting and treating these specific HPV types, as identified in local cervical cancer cases [38], are shown to have a notable impact on preventing a significant proportion of the associated cervical cancer burden. This distinction emphasizes the importance of tailoring screening approaches to the specific context of cervical cancer etiology, enhancing the precision and effectiveness of preventive measures within the studied population.

Table 5 summarizes our key findings, encompassing the observed mRNA positivity rates, CIN3+ sensitivity, and the number of mRNA-positive cases needed to address one CIN3+ case, serving as a metric for screening effectiveness. Additionally, the table presents the estimated proportion of preventable cervical cancer cases based on treating the identified HPV types in local cervical cancer cases [38]. Our analysis revealed that, on average, detecting and treating two mRNA-positive cases of high-risk HPV was necessary to treat one existing CIN3+ case. The proportion of prevented cancers by type exhibited minimal variation between HIV-negative and HIV-positive women, suggesting consistent potential for cervical cancer prevention across these groups (Table 5).

Using a 4-type mRNA test (types 16, 18, 45, and 35) for screening in a population with a high prevalence of CIN3 cases required treating 30% of HIV-positive and 10.7% of HIV-negative individuals (representing 20% of the overall study population). This approach identified 41.1% of prevalent CIN3 cases and had the potential to prevent 77.6% (95% CI: 71.2–84.0) of the associated cervical cancer burden.

Similarly, a 6-type (16, 18, 45, 31, 33, 35) mRNA primary screening test necessitated treating nearly 25% of the study population, detecting 62% of all prevalent CIN3 cases, and exhibited the potential to prevent 85% of cervical cancers (95% CI: 79.6–90.6), with an even distribution among HIV-positive and HIV-negative women.

## 4. Discussion

### 4.1. HPV Prevalence and Testing Approaches

The transition from conventional to liquid-based cytology and the introduction of HPV testing in South Africa’s laboratories marks a significant step forward. Nevertheless, the prevalence of HPV in the general female population varies markedly, ranging from 76.1% overall to 37.1% for the types included in the Gardasil-9 vaccine [41]. From the Cape Town region, other studies have reported twice as high HPV prevalence rates among HIV-positive women (52.4%, 40.6%, 45.5%) compared to HIV-negative women (20.8%, 21.4%, 20.3%) [42,43,44]. This high prevalence of HPV, particularly among HIV-positive individuals, poses challenges for employing HPV DNA tests as a primary screening method, as their implementation would necessitate substantial healthcare resources for follow-up or treatment in a system that is already strained by the demands of cytological screening.

In contrast to HPV DNA tests, which detect both transient and persistent infections, HPV mRNA tests primarily detect persistent infections where viral DNA may not be integrated into the host cell’s DNA [35]. The mRNA tests employed in our study specifically target the viral oncogenes E6 and E7 of high-risk HPV types. The expression of these oncogenes leads to the degradation of tumor suppression genes (p53/retinoblastoma protein), disrupting cell cycle regulation and potentially driving the progression of advanced CIN grades and ultimately invasive cervical cancer [45].

Our study’s findings demonstrated an expected increase in the HPV mRNA positivity rate by increasing number of types involved, with more than a two-fold rise observed from the 3-type (8.8%) to the 8-type (19.0%) mRNA test combination. This increase was consistently observed among HIV-positive women, who exhibited a positivity rate 2–3 times higher than that of HIV-negative women, in line with previous reports [42,43,44]. Furthermore, our data revealed that the prevalence of CIN3+ was twice as high among HIV-positive women compared to their HIV-negative counterparts. Histological diagnosis indicated that the positivity rate across various mRNA combinations was slightly higher among HIV-positive women, although the differences were not statistically significant. The majority of histologically confirmed cervical cancers (11 out of 12) were found to be positive for mRNA of the top 5 types.

### 4.2. Sensitivity, Specificity, and Predictive Values

The sensitivity of HPV mRNA test combinations in detecting CIN3+ disease displayed a significant increase with the inclusion of additional HPV types, ranging from 46.1% (3 types) to 84.3% (8 types) among HIV-positive women. Interestingly, Snyman et al. reported CIN3+ sensitivity at 77.1% for the HPV DNA test including 13 types (Hybrid capture 2 (HC2), Qiagen, Germantown, MD, USA) among HIV-positive women, who also were recruited within the larger Diagnostic Vaccine and Cervical Cancer Screen (DiaVACCS) initiative [44]. The sensitivity enhancement was more pronounced among HIV-positive women than their HIV-negative counterparts. Conversely, as the number of mRNA types expanded, there was a corresponding decline in specificity, with a more moderate reduction observed among HIV-negative women (from 94.5% to 88.0%). Markedly, the positive predictive values for various mRNA combinations exhibited consistency irrespective of HPV–HIV status. It is worth noting that the observed 50% PPV for CIN3+ contrasts with the reported 3.8% PPV for CIN3 in DNA testing among HIV-positive individuals [42] and 30.3% as reported by Snyman et al. [44]. This stresses the advantages of mRNA testing in high-prevalence HPV populations. Notably, the low number needed to treat mRNA-positive cases to manage a single CIN3+ case, ranging from two to one for any combination of mRNA types, highlights the efficiency and clinical impact of mRNA-based screening.

### 4.3. Cervical Cancer Prevention Potential

Based on the 12 cases in this study, the sensitivity for invasive carcinoma was 91.7% for tests including the top 5, 6, 7, and 8 types, emphasizing the robust performance of mRNA testing even using a limited number of types.

The study findings accentuate the significant potential for cervical cancer prevention through targeted screening approaches using HPV mRNA tests. Both the 4-type (16, 18, 35, 45) and 6-type (16, 18, 31, 33, 35, 45) mRNA combinations demonstrated the capacity to prevent approximately 78% and 85% of cervical cancer cases, respectively, by identifying a substantial portion of CIN3 disease cases by treating 20% to 25% of the study population. These results highlight the effectiveness of such strategies, offering the potential for a substantial reduction in cervical cancer burden, particularly in high-risk populations. It is important to note that the composition of our study population, with a higher proportion of HIV-positive women, may impact the generalizability of these findings to the broader South African population.

### 4.4. Considerations and Future Directions

Our study population consisted of relatively unscreened women (without previous screening or with a lapse of five years since their last screen) but was enriched for cervical disease. Moreover, the proportion of HIV-positive women (47.5%) in our study was twice that of the general female population reported in 2020 (42). Despite these discrepancies, the positivity rate for any mRNA combination remained relatively low in this high-risk population, indicating the potential utility of mRNA tests in such settings.

Demonstrably, HIV-positive women are diagnosed with ICC approximately 15 years earlier in life than HIV-negative women [46]. Due to reduced immune competence, despite ARV therapy, lesions tend to progress faster than in HIV-negative women. ARV therapy improves life expectancy and ensures more time to treat CIN lesions, in time to prevent progress to ICC.

It is important to recognize that while treating CIN3 lesions is crucial for preventing cervical cancer, not all CIN3 lesions progress to invasive cancer [47,48,49]. CIN3 lesions may be caused by HPV types with low, if any, potential for ICC progression. The validity of the classification of oncogenic HPV types based solely on their presence in cervical pre-cancer and cancer tissue has been questioned [49,50,51]. Treating all CIN3 lesions without considering the respective HPV type’s potential for ICC progression may lead to unnecessary overtreatment, straining already limited healthcare resources.

This study demonstrated that by treating 25% of the study population screened with the 6-type mRNA test, the potential to prevent 85% of ICC cases is possible. The compelling results from our study underscore the critical importance of further research, including extensive controlled studies, to validate and refine the proposed “screen and treat” strategy. Continued investigation and optimization of such approaches hold the potential to revolutionize cervical cancer prevention, particularly in regions grappling with limited resources.

### 4.5. Strengths and Limitations

Our study benefited from a robust design, ensuring material representative for the South African general population [35]. Most screening guidelines are based on HPV and disease prevalence data which originated in areas with considerably different demographics and HIV status, with a large impact on HPV prevalence. However, local data is critical to tailor effective screening strategies. Having histological evaluation and mRNA test results for all included participants enabled a thorough evaluation of test performance. Additionally, the well-characterized study population, being a multicentric study with thorough staff training, strengthened our study. The presented baseline results, with the inclusion of all WHO screening options to evaluate the number of screen-positive provide a valuable reference for future decision-making [36,52,53].

The most important limitation of the current study is that the study population does not totally reflect the South African screening population and should therefore not be extrapolated to the general population. Firstly, the HIV prevalence and ARV status do not represent that of the South African screening population, in that HIV-positive women are intentionally overrepresented to have a sufficiently large HIV population for accurate sub-group analysis. Secondly, the study group was enriched for cervical disease when all women without histology were excluded from the calculation of test performance to ensure accuracy.

Among the 1001 women mRNA tested, 291 women were excluded because of a lack of histology, unknown HIV status, or having invalid test results. Among the 251 participants who were excluded due to the absence of histology, 96% were mRNA-negative and 97% had normal cytology, indicating the missed disease burden to be relatively low. Whilst the first publications from this cohort presented data for all women and used multiple imputations to estimate verification bias-adjusted histology values, in this analysis we elected to calculate test performance purely on known histology-proven disease status, to ensure robust test performance outcomes.

## 5. Conclusions

In conclusion, the test-and-treat approach emerges as a promising strategy for cervical cancer prevention in resource-constrained settings. Our study sheds light on the effectiveness of this approach, particularly in the context of a population characterized by a high prevalence of HIV, HPV, and CIN3+. Utilizing either a 4-type or 6-type mRNA test in unscreened or under-screened women presents a compelling opportunity to detect the vast majority of women at high risk to prevent progression to invasive cervical cancer. We consider the calculated number of women needed to treat in order to detect and treat a single case of CIN3+ in this study as very acceptable. Treatment capacity should probably be used to determine the treatment threshold and in this case the number of HPV types in the test. Ultimately, through strategic implementation and continuous advancement, we can aspire to mitigate the impact of cervical cancer and improve the well-being of countless women in vulnerable populations.

## Figures and Tables

**Figure 1 cancers-15-05453-f001:**
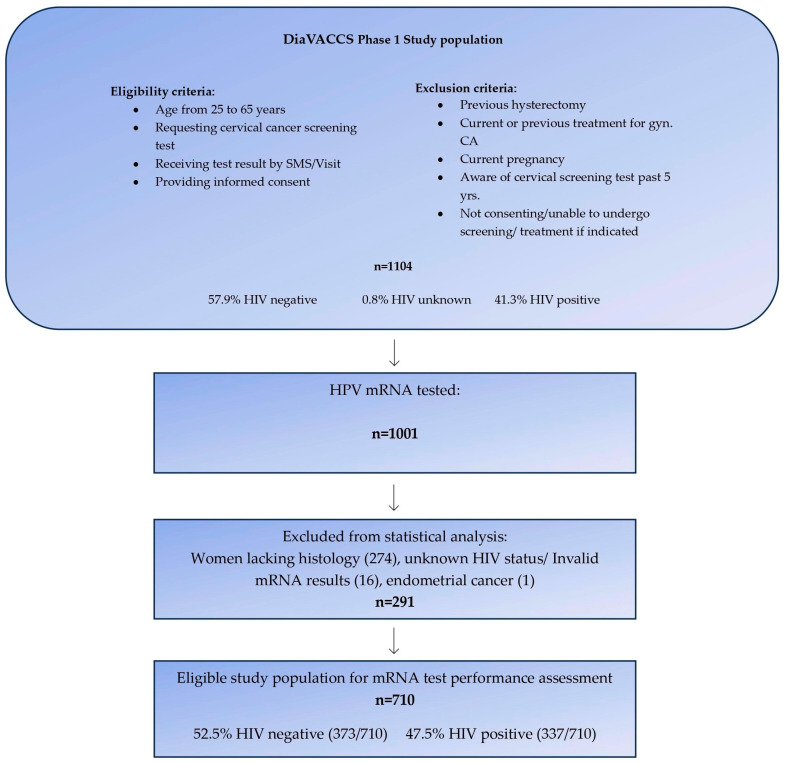
Overview selection of study population.

**Table 1 cancers-15-05453-t001:** HPV mRNA type frequency.

mRNA HPV Type	(n)	(%)
16	48	6.8
45	33	4.6
58	29	4.1
18	27	3.8
52	27	3.8
31	24	3.3
35	21	3.0
33	15	2.1
Negative	486	68.5
Overall positive	224	31.5

**Table 2 cancers-15-05453-t002:** Age, the positivity rate for mRNA combinations, and most severe histology, all by HIV status.

Characteristics	HIV Neg.	HIV Pos.	Total	*p*-Value
Women (n)	373	337	710	
Age n (years)				
25–39	181 (48.5)	171 (50.7)	352 (49.6)	*p* = 0.60
40–65	192 (51.5)	166 (49.3)	358 (50.4)	
mRNA HPV types n (%)				
16, 18, 45	33 (8.8)	75 (22.3)	108 (15.2)	*p* < 0.01
16, 18, 45, 35	40 (10.7)	101 (30.0)	141 (19.9)	
16, 18, 45, 31, 33	46 (12.3)	101 (30.0)	147 (20.7)	
16, 18, 45, 31, 33, 35	52 (13.9)	123 (36.5)	175 (24.6)	
16, 18, 45, 31, 33, 52, 58	67 (18.0)	136 (40.4)	203 (28.6)	
16, 18, 45, 31, 33, 52, 58, 35	71 (19.0)	153 (45.4)	224 (31.5)	
Most severe histology n (%)				
Normal	139 (37.3)	98 (29.1)	237 (33.4)	
CIN1	123 (33.0)	72 (21.4)	195 (27.5)	
CIN2	63 (16.9)	78 (23.1)	141 (19.9)	*p* < 0.01 *
CIN3	44 (11.8)	81 (24.0)	125 (17.6)	*p* < 0.01 **
ICC	4 (1.1)	8 (2.4)	12 (1.7)	

CIN = Cervical Intraepithelial Neoplasia, ICC = Invasive Cervical Cancer; * CIN2+/<CIN2, ** CIN3+/<CIN3.

**Table 3 cancers-15-05453-t003:** The detection rate of mRNA HPV genotype combinations by most severe histology across HIV status.

Histology	Normal		CIN1		CIN2		CIN3		ICC	
HIV Status	Neg.	Pos.	Neg.	Pos.	Neg.	Pos.	Neg.	Pos.	Neg.	Pos.
Women (N)	139	98	123	72	63	78	44	81	4	8
mRNA HPV+ (%)										
16, 18, 45	2.2	6.1	4.1	2.8	15.9	33.3	27.3	44.4	75	62.5
16, 18, 45, 35	2.9	9.2	4.1	4.2	20.6	47.4	34.1	56.8	75	75
16, 18, 45, 31, 33	3.6	7.1	5.7	12.5	17.5	42.3	45.5	54.3	75	100
16, 18, 45, 31, 33, 35	4.3	10.2	5.7	13.9	22.2	55.1	50	64.2	75	100
16, 18, 45, 31, 33, 52, 58	6.5	10.2	9.8	18.1	23.8	56.4	63.6	75.3	75	100
16, 18, 45, 31, 33, 52, 58, 35	7.2	13.3	9.8	19.4	27	65.4	65.9	82.7	75	100

**Table 4 cancers-15-05453-t004:** Sensitivity, specificity, positive and negative predictive value for CIN3+ by HIV status and for the total group.

	Sensitivity			Specificity			PPV			NPV		
HIV Status	Neg.	Pos.	Total	Neg.	Pos.	Total	Neg.	Pos.	Total	Neg.	Pos.	Total
mRNA HPV+	%	%	%	%	%	%	%	%	%	%	%	%
16, 18, 45	31.3	46.1	40.9	94.5	86.3	90.9	45.5	54.7	51.9	90.3	81.7	86.5
16, 18, 45, 35	37.5	58.4	41.1	93.2	80.2	87.6	45.0	51.5	49.6	91.0	84.3	88.2
16, 18, 45, 31, 33	47.5	58.4	54.7	92.9	80.2	87.4	50.0	51.5	51.0	92.4	84.3	89.0
16, 18, 45, 31, 33, 35	52.1	67.4	62.0	91.7	74.6	84.3	48.1	48.8	48.6	92.8	86.4	90.3
16, 18, 45, 31, 33, 52, 58	64.6	77.5	73.0	88.9	73.0	82.0	46.3	50.7	49.3	94.4	90.0	92.7
16, 18, 45, 31, 33, 52, 58, 35	66.6	84.3	78.1	88.0	68.5	79.6	45.1	49.0	47.8	94.7	92.4	93.8

PPV = Positive Predictive Value, NPV = Negative Predictive Value.

**Table 5 cancers-15-05453-t005:** mRNA positivity rate, CIN3+ sensitivity, number of mRNA positive cases needed to treat to address one existing CIN3+ case, estimated proportion of cervical cancers prevented by detection and treatment of CIN3+ by the number of mRNA HPV types included in the test algorithm. Note that for the HPV-type combinations presented, HPV 35 is included in following combinations (4, 6, and 8), whilst not included in combinations (3, 5, and 7).

	Positivity Rate mRNA			Sensitivity CIN3+			NNT * to Address One Case CIN3+			Estimated % CC Prevented ** [38]		
HIV Status	Neg.	Pos.	Total	Neg.	Pos.	Total	Neg.	Pos.	Total	Neg.	Pos.	Total
Women (N)	373	337	710	48	89	137	NA	NA	NA	96	65	161
mRNA HPV+	%	%	%	%	%	%	%	%	%	%	%	%
16, 18, 45	8.8	22.3	15.2	31.3	46.1	40.9	2.2	1.8	1.9	66.7	70.8	68.3
16, 18, 45, 35	10.7	30.0	19.9	37.5	58.4	41.1	2.2	1.9	2.0	78.1	76.9	77.6
16, 18, 45, 31, 33	12.3	30.0	20.7	47.5	58.4	54.7	2.0	1.9	2.0	72.9	80.0	75.8
16, 18, 45, 31, 33, 35	13.9	36.5	24.6	52.1	67.4	62.0	2.1	2.1	2.1	84.4	86.2	85.1
16, 18, 45, 31, 33, 52, 58	18.0	40.4	28.6	64.6	77.5	73.0	2.2	2.0	2.0	78.1	83.1	80.1
16, 18, 45, 31, 33, 52, 58, 35	19.0	45.4	31.5	66.6	84.3	78.1	2.2	2.0	2.1	89.6	89.2	89.4

* NNT= Number Needed to Treat to address one case of CIN3+; ** Estimated proportion of cervical cancers prevented by detection and treatment of CIN3+ cases caused by specific high-risk HPV types.

## Data Availability

The data can be shared up on request.

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
