# Peer review of "Enhancing Cervical Cancer Prevention in South African Women: Primary HPV mRNA Screening with Different Genotype Combinations"

_cancers, 2023, doi:10.3390/cancers15225453_

Round 1

Reviewer 1 Report

Comments and Suggestions for Authors

This study explored the potential preventive value of a "test-and-treat" strategy using mRNA HPV tests to impact cervical cancer prevention in unscreened or under-screened South African women with a high-prevalence HIV population. The authors used combinations of three to eight HPV types in the mRNA tests for detecting and treating cervical cancer.  Based on this work, the 6-type HPV mRNA test had the best performance; and the proposed “test-and-treat” approach proved to be a very promising strategy for cervical cancer prevention in resource-constrained settings.  The work is well-designed, the method is sound, the results are well-presented, and the manuscript is well-written.  If validated and optimized, this "screen and treat" strategy could revolutionize cervical cancer prevention, particularly in high-risk and vulnerable populations.

Minor suggestions:

The first paragraph in the Discussion, "Challenges in Cervical Cancer Screening", could move to the Introduction, as this is one of the important points and significance of this work.  

Author Response

Reviewer 1:

This study explored the potential preventive value of a "test-and-treat" strategy using mRNA HPV tests to impact cervical cancer prevention in unscreened or under-screened South African women with a high-prevalence HIV population. The authors used combinations of three to eight HPV types in the mRNA tests for detecting and treating cervical cancer.  Based on this work, the 6-type HPV mRNA test had the best performance; and the proposed “test-and-treat” approach proved to be a very promising strategy for cervical cancer prevention in resource-constrained settings.  The work is well-designed, the method is sound, the results are well-presented, and the manuscript is well-written.  If validated and optimized, this "screen and treat" strategy could revolutionize cervical cancer prevention, particularly in high-risk and vulnerable populations.

Minor suggestions:

The first paragraph in the Discussion, "Challenges in Cervical Cancer Screening", could move to the Introduction, as this is one of the important points and significance of this work.  

Our response:

Dear Reviewer,

We sincerely appreciate your time and effort in reviewing our manuscript. We are pleased to receive your feedback and are grateful for your acknowledgment of the relevance of our study's subject, as well as your positive assessment of our work method and results.

In response to your suggestion regarding the section “Challenges in Cervical Cancer Screening, we have moved this part to the Introduction (L73-81).

Reviewer 2 Report

Comments and Suggestions for Authors

In the manuscript by Sorbye et al the authors describe a robust study of HPV mRNA assessment in cervical cancer screening in a challenging population. The study was performed on a significant number of participants from which less reliable cases without histology were excluded. The methodology is robust and conclusions valid.

There are some minor problems with the manuscript that could be further improved.

P3 L130 while the detailed data was previously published, consider at least briefly listing the overall prevalence of the most common 6 HPV types so that the combinations later used throughout the manuscript can readily be evaluated in the context of prevalence of those types. This would make the current manuscript more standalone.

P3 L135-136 is it justified to take 2 biopsies in women not having lesions (negative screening recall - or subsequently blind concurrent biopsies)? Were alternative methods of identifying true negatives considered?

P3 L 140 It is a bit unclear what the percentages 91.7 and 49.1 are based on. How do they relate to the n=1104 / 1001 / 710 sample sizes mentioned later in the manuscript. This is further complicated by „screen-positive“ referring to cytology and/or HPV mRNA detection

P4 L147 authors explicitly mention punch and treatment biopsies. Does the punch biopsy refer to biopsies in women where lesions were not apparent on colposcopy (P3 L135) while treatment biopsy refers to targeted biopsies (p3 L134)?  Do the results contain LEETZ histology material or only biopsies? Please include this explicitly as well as make the text more streamlined in terminology if possible

P4 L150 possibly some name or reference for the HIV test would be warranted

P4 L153 does the “mRNA types” refer to HIV mRNA or HPV mRNA typing?

P5 L197 Typo „identification number NCT0295603” should be “NCT02956031”. https://clinicaltrials.gov/study/NCT02956031

P5 L 201   the description of exclusion criteria and final sample numbers might be better suited for the materials and methods section.

It is also unclear from the text up to this point what would be the reason ~10% of women did not have mRNA test results available? Only for 16 women it is implied their HIV or HPV tests were invalid

P5L203 possibly better connection between women lacking histology (n=274) at this point and previous study population descriptions would be helpful.

P5 L203   since the study originally included women up to march 2020, is there some followup available in the last 3 years for the 274 women missing histology that would decrease the number of women excluded for this reason?

Do the results change significantly if the subset of the 274 women without cervical abnormalities and without histology results are treated as “Normal” to avoid any exclusion bias?

P6 L 208 Figure 1. There is some mismatch between the inclusion/exclusion criteria listed in the figure and the respective manuscript text at P3 L118-124. What would be hesitancy to undergo screening?

P7 L2018 there are some minor rounding problems with the percentages (ie. 33.4+2.5+19.9+17.6+1.7 =100.1%)

For Table1 consider including both the number and percent for data. There appears to be enough room on the page to include this and it might be more informative

P8 L251  taking only a brief look at the text  might be confusing. Please consider clarifying or elaborating on the results of Rad et al study at this point so that the distinction of 159 cases (P8 L251) and 161 cases (P9L278) is more easily understood without having to study the reference in detail.

P8 L248 there is a formatting problem with borders of table 3 (border below PPV and NPV columns is missing)

P9 Table 4 highlights a slight issue with the data on HPV types as presented. The authors throughout the manuscript imply the major difference between rows of combinations as presented in tables 1-4 is the number of types in the combined result. However due to the composition of the tests employed (16, 18, 45, 35) and (16, 18, 31, 33, 45, 52, 58) there is also a qualitative difference in some of the rows which might be best to avoid. Namely row combination of 4 types (16,18,45,35) is found in 77.6% of cancer cases in the hospital dataset while that of 5 types (16,18,45,31,33) suddenly has less prevalence being observed in 75.8%. Some readers might miss the fact that 4 type combination included HPV 35 while the 5 type combination does not include the same type 35. It seems that HPV35 is markedly increased in HIV+ cases since the combinations including this type show marked increase in positivity rate.

On the other hand, the above comment might require too much time and effort to address while changing the groupings would likely not affect the conclusions and the authors can skip it if they wish

Author Response

See attached pdf file
